# Analysis of the Corrosion Process with the Application of the Novel Type of Coupon Installation

Daniel Musik [1], Krzysztof Wójcik [1], Małgorzata Sekuła-Wybańska [1], Maciej Konopacki [2] and Rafał Rakoczy [2,*]

[1] ESC Global sp. z o.o Słoneczny Sad 4F, 72-002 Dołuje, Poland
[2] Faculty of Chemical Technology and Engineering, West Pomeranian University of Technology in Szczecin, 42 Piastów Avenue, 71-065 Szczecin, Poland
* Correspondence: rrakoczy@zut.edu.pl

**Abstract:** The corrosion process leads to high power consumption, high maintenance costs and the loss of commercial income during downtime in various branches of industry. The proper methods to measure and forecast the corrosion process would help intervene in process production where corrosion is a common phenomenon. Therefore, the main aim of this experimental study is to improve the widely used corrosion monitoring methods with corrosion coupons. As part of this work, the installation for testing corrosion process under controlled conditions and with the application of mild steel coupons is proposed. The measurement concept is to install the coupons in a stream with the corrosion liquid (these conditions should be controlled). The numerical simulations of the fluid flow in the coupon installation were carried out, and the obtained results in the form of a velocity map allowed us to propose the placement of the coupons in the tested installation in such a way that the flowing liquid evenly washed the coupon surface. The developed coupon installation was tested for aggressive corrosive conditions, which were assessed using the water stability indices (Langelier Saturation Index and Ryznar stability index). Moreover, the inductively coupled plasma optical emission spectroscopy analysis characterised the liquid samples from the tested coupon installation. The corrosion process for the applied conditions was defined based on the corrosion rate of the tested coupons. This process was also confirmed by obtaining the Raman spectrum for the used corrosion coupons. The obtained investigation contributes significantly by developing the novel coupon installation and demonstrating the procedure for testing the corrosion process with the application of coupons. This setup and method might be successfully applied for accelerated laboratory tests.

**Keywords:** coupon installation; corrosion; controlled conditions of corrosion; water stability indices



## 1. Introduction

The rapid technological progress in all industries has resulted in the appearance of more complicated materials and various technological installations. It should be noted that construction materials (mainly metals) interact with the surrounding environment. This process is called corrosion, one of the most critical aspects in the design of any installations or systems. The corrosion process causes uncontrolled and unforeseen failures in laboratory or industrial setups. The corrosion process is an issue that has always been accompanied by the application of metals in aggressive environments. The dynamic growth of the chemical industry is related not only to the demand for new apparatuses or plants, but also to the need for the control and maintenance of setups' parts.

The main aim of corrosion protection is to reduce the corrosion rate to an acceptable level. The corrosion protection methods should be introduced at the initial stages of the design process of the apparatus. Firstly, attention should be paid to selecting suitable construction materials, considering the contact of constructional steel with the other materials. Secondly, the corrosive environment should be analysed. It should be noted that

corrosion inhibitors might be introduced to this environment to limit corrosion. Moreover, the application of methods of protection against corrosion is recommended, such as the application of inhibitors, electrochemical protection, coatings, and environmental modifications [1].

From a practical standpoint, we have two primary corrosion test methods:, laboratory tests and tests in natural conditions. The first group of methods (laboratory tests) are corrosion tests carried out in artificially created conditions, which imitate natural conditions. These tests might also be realised by the changed conditions to obtain the acceleration of the corrosion processes compared to the operating conditions. The corrosion tests in natural conditions are carried out in the atmosphere, soil, and seawater using specially prepared samples [2]. These tests are directly performed on industrial devices and the obtained results help produce suitable materials or protection methods for the specific processes. It should be noted that the corrosion tests might be carried out in laboratory conditions [3].

The model tests are carried out using laboratory setups in which the conditions existing on the industrial scale are imitated. The model tests on the laboratory scale are realised using specially prepared samples (coupons) placed in the liquids, the industrial solutions or the prepared solutions in the laboratories. The apparatus used for these tests in the liquids or solutions should have a temperature-stabilising system to maintain and control the assumed process temperature. The samples (coupons) should be completely submerged in the liquid or solution. In the case of accelerated laboratory tests, one or more corrosive factors are strengthened (e.g., temperature, relative humidity, moisture condensation, and the concentration of corrosive components such as $SO_2$, $H_2S$, ammonia, acids, and chlorides). The corrosion process under these conditions is faster than the process carried out in operating conditions.

It is now well established from various studies that water stability indices are used to assess water corrosivity [4]. The estimation of these parameters is allowed to define the scaling and corrosive nature of the water used in the laboratory setups. The Langelier Saturation Index (LSI) and the Ryznar Stability Index (RSI) were introduced to infer the scale-forming potential of an aqueous solution from its composition [5]. The LSI provides a measure of the stability of water for its degree of $CaCO_3$ saturation [6]. This parameter can estimate the affinity of water to dissolve or precipitate $CaCO_3$, which is the main factor influencing water's corrosivity [7]. The RSI used the LSI as a component in a formula to improve the accuracy in predicting water scaling or corrosion tendencies [7]. This index provides information about the scale thickness observed in water systems to the water chemistry [5].

The laboratory scale's corrosion tests are carried out using specially designed chambers or apparatus. Introducing corrosion samples (coupons) into various types of installations is one of the most frequently used corrosion monitoring methods. The coupons are inserted into the installation for a specified period. After this process, the coupons are removed from the experimental setup, tested and weighed. The typical examples of the experimental setups used in the corrosion tests are presented in Table 1.

The main aim of this paper is to evaluate the novel type of coupon installation which might be used to measure and forecast the corrosion process. The proposed installation would help in intervention in process production where corrosion is a common phenomenon. The coupon installation is tested under controlled conditions (application of 1M NaOH). It should be noted that preventing the corrosion process allows economic and energy savings. Therefore, this experimental study attempts to improve the widely used corrosion monitoring methods with corrosion coupons and a novel coupon installation.



**Table 1.** The typical examples of the experimental setups used in the corrosion tests.

| Ref. | Description |
|---|---|
| [8] | The installation for testing the corrosion process is placed in an autoclave (a hermetically sealed tank in which the chemical process under the created conditions is carried out). These conditions are created considering the tested parameter, i.e., temperature, total pressure, gas mixture, brine, time, types of coupons and rotational speed. The coupons are connected to the rod and placed in the setup's central part. The installation is used to test the resistance of materials to corrosion, which is used in the production of drilling pipes and covers. |
| [9] | The installation for testing corrosion of FLiNaK salt (46.5% LiF-11.5% NaF-42% KF (mol%)) is presented. This molten salt is proposed as a coolant for reactors and a heat transfer carrier for nuclear reactor plants. The installation contained a 316L steel crucible with steel and graphite coupons. |
| [10] | The corrosion installation consisted of a network of parallel pipes connected via pipe fitting (bends, t-joints, valves etc.). The mass flow rate, pressure, pH, temperature, and humidity sensors are used to monitor corrosion conditions. |
| [11] | Using different carrier gasses, a dynamic flow-through system was assembled to allow for dynamic iodine exposures on hanging substrates. The metal coupons were tested in the setup by monitoring temperature, iodine concentration, flow rate, atmosphere, and relative humidity. |

## 2. Materials and Methods

### 2.1. Experimental Setup

A novel type of coupon installation (patent application N P.440109; WIPO ST 10/C PL440109) is presented in Figure 1. The coupon installation might be divided into three modules. The first module consists of a tank (1) for the working liquid (corrosive conditions of interest). This tank is equipped with protection against low liquid levels (3). To control the temperature during the tests, the setup is equipped with a heater (6). The main element of the second module is the heater controller (2) connected to the temperature sensor (5). The additional element of this system is protection against exceeding the maximum temperature of the liquid in the tank (4). The tank (1) is equipped with a system of liquid circulating: the circulating pump (7), filter (8), three-way valve (9), a plate heat exchanger (10), and the vent valve for the circulating system (11). This system is used to cool the working liquid in the coupon installation. The third module consists of the piping system for online corrosion monitoring (zig-zag system). This system is made of transparent polycarbonate, allowing it to observe the progress of the corrosion process on the tested coupon samples (13). It should be noticed that these coupons might be placed into the working liquid parallel to the direction of fluid flow. In addition, it is possible to monitor the corrosion process on the coupons directed in and against the direction of the fluid flow. The coupon installation comprises the rotameter (12) and the vent valve for the zig-zag system (14).

Figure 2 shows the main dimensions of the zig-zag system. It should be noted that the proposed installation works periodically. The zig-zag system is directly connected to the tanks. It is also possible to install this system as a bypass system to existing piping.

The tested setup is also equipped with adjustable coupon holders (see Figure 3). The application of this holder is allowed to adjust the position of the coupon in the flow flowing through the zig-zag system.

### 2.2. Temperature Conditions

The developed coupon installation is allowed to set and monitor the flowing fluid temperature through the setup. The experimental procedure was carried out with a fluid temperature of 40 °C.

### 2.3. Hydrodynamic Conditions

In the case of testing the coupon system, it was possible to set three gears of the circulation pump, which forced the liquid to circulate in the coupon installation. The linear velocity of the fluid in the tested coupon installation was equal to 0.71, 1.29, and 1.46 m·s$^{-1}$. The hydrodynamic conditions in this system might be defined by using the dimensionless Reynolds number, as follows:

$$\mathrm{Re} = \frac{w\,d}{\nu} \tag{1}$$

where:

$w$—linear velocity of the liquid, m·s$^{-1}$;

$d$—diameter of the pipe in the coupon installation (in zig-zag system), m;

$v$—viscosity of fluid, m$^2$·s$^{-1}$.

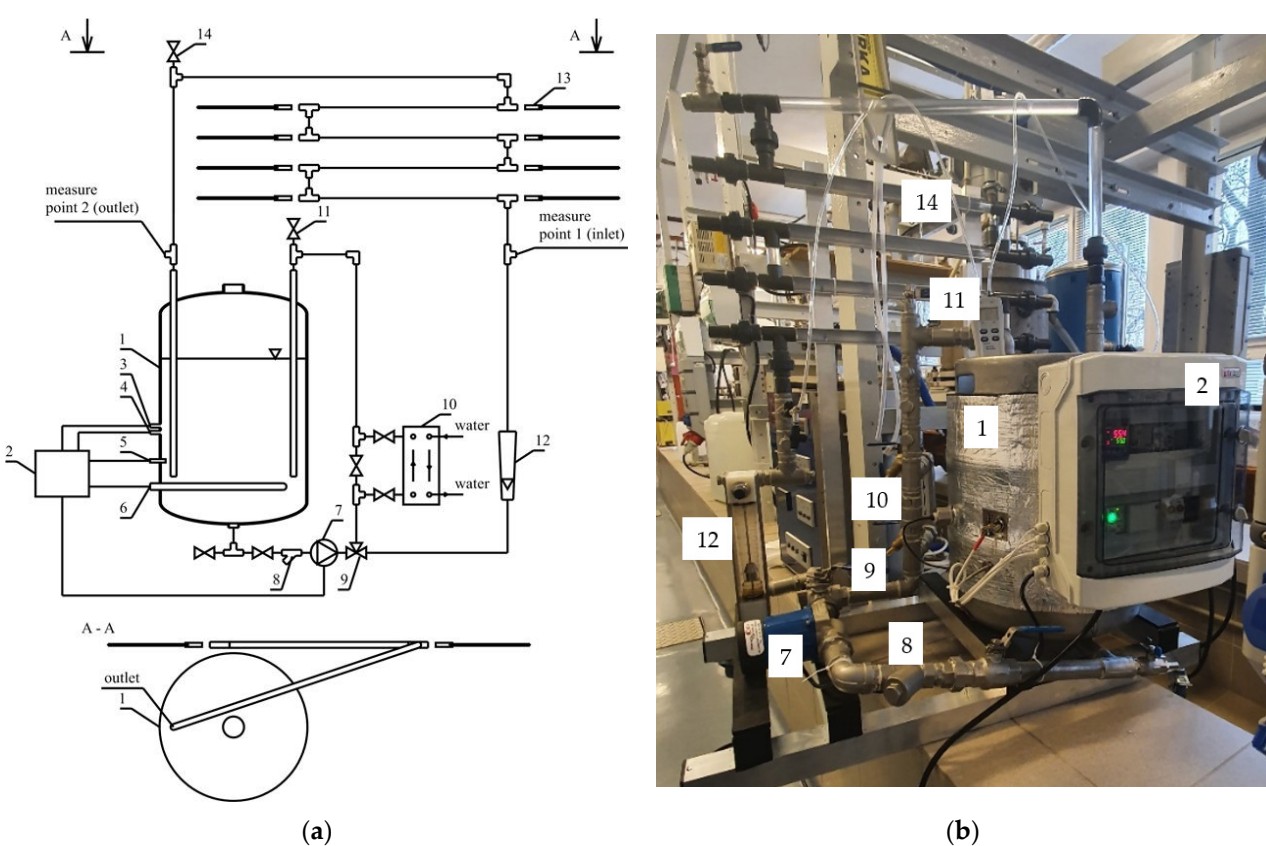

| (**a**) | (**b**) |

**Figure 1.** Schematic of the experimental setup (**a**) and installation photo (**b**): 1—tank; 2—heater controller; 3—protection against low liquid level in tank; 4—protection against exceeding the maximum temperature of the liquid in tank (maximum operating temperature is 60 °C); 5—temperature sensor; 6—heater; 7—circulation pump; 8—filter; 9—three-way valve; 10—plate heat exchanger (for cooling); 11—vent valve for circulating system; 12—rotameter; 13—tested coupon sample; 14—vent valve for circulating system with coupons (zig-zag system).

### 2.4. Investigation of Pressure Drop

The pressure drop across the coupon installation was measured employing the digital pressure gauge CPG1500 (WIKA Poland S.A.). All experiments were performed systematically with varying linear velocities of fluid (0.71, 1.29, and 1.46 m·s$^{-1}$). The value of this parameter was obtained for the installation without (empty) and with coupons.

### 2.5. Numerical Simulaiton

To analyse the fluid flow in the tested coupon installation, the numerical simulations of CFD were performed using ANSYS Software. Two various 3-D models of the configuration of the coupon installation were analysed: the installation without (empty) and with coupons. A hexagonal dominant numerical grid was created for each geometry, mainly by the sweep method, which results in a high-quality mesh. The typical view of numerical mesh is presented in Figure 4.

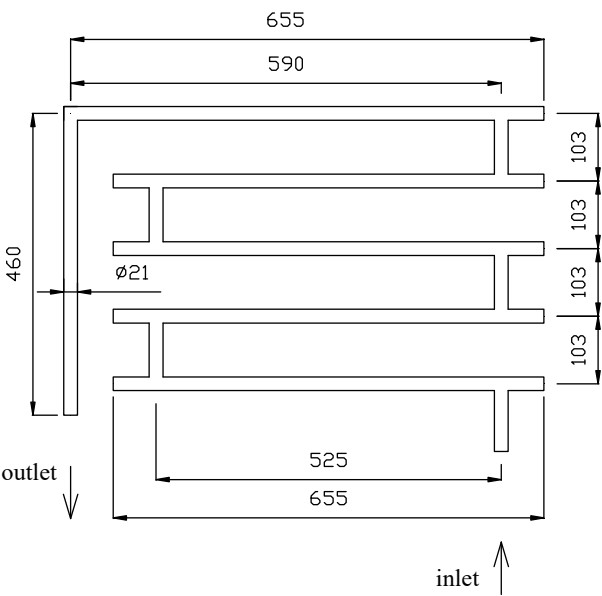

**Figure 2.** The main dimensions of the zig-zag system used in the novel type of coupon installation.

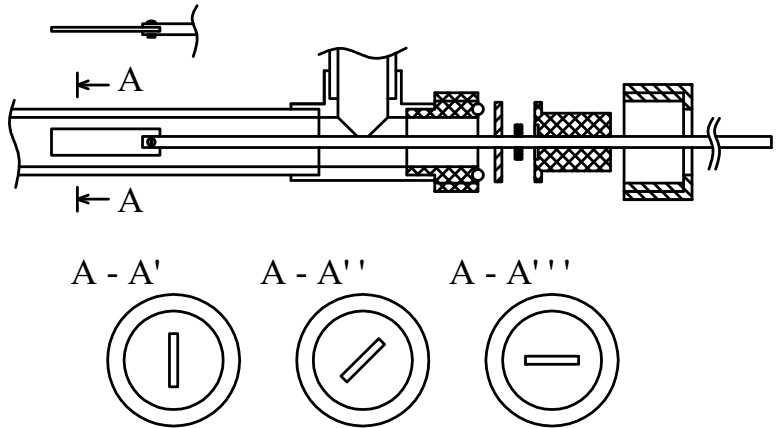

**Figure 3.** The coupon holder is used in the novel type of coupon installation.

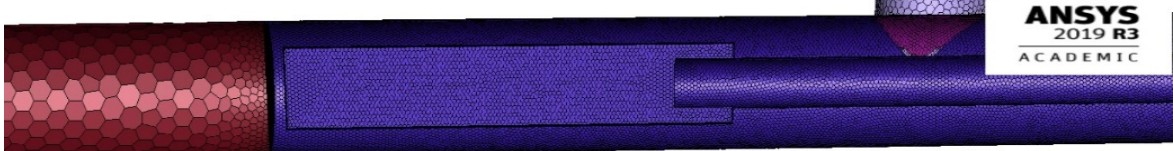

**Figure 4.** The view of numerical mesh is applied for the numerical analysis.

The properties of the used numerical grids that were utilised in CFD analysis are presented in Table 2.

**Table 2.** Characteristics of the numerical grids used in the numerical analysis.

| Type of Coupon Installation | Parameters | | | |
|---|---|---|---|---|
| | Number of Cells | Number of Nodes | Skewness | Orthogonal Quality |
| Without coupons (empty) | 90 601 | 98 584 | 0.177 | 0.953 |
| With coupons | 679 776 | 187 412 | 0.230 | 0.779 |

The CFD simulations were performed using ANSYS Fluent solver. The k-omega SST model for fluid turbulence calculation was employed with default coefficient values. Moreover, the typical boundary conditions were used (velocity inlet and pressure outlet). The three values of the linear velocity of fluid were used as the velocity inlets. At the outlet, the relative pressure of 0 Pa was specified, corresponding to the atmospheric pressure outside the system. The simulations were performed until the $10^{-5}$ residual level was reached. As a result, contour plots of fluid velocity distribution in the system's vertical cross-section were created. A fluid pressure drop was also estimated based on inlet and outlet pressure values for each analysed case.

*2.6. Experimental Procedure*

The coupon installation was tested by using aggressive corrosion conditions. All corrosion tests were performed in demi water (pH 5.5–7; conductivity < 0.1 μS·cm$^{-1}$; heavy metals (pb) < 0.1 ppm; aluminium < 0.05 mg·L$^{-1}$; barium < 0.01 mg·L$^{-1}$; calcium < 0.01 mg·L$^{-1}$; cadmium < 0.01 mg·L$^{-1}$; chromium < 0.01 mg·L$^{-1}$; copper < 0.01 mg·L$^{-1}$; iron < 0.01 mg·L$^{-1}$; potassium < 0.01 mg·L$^{-1}$; magnesium < 0.01 mg·L$^{-1}$; manganese < 0.01 mg·L$^{-1}$; molybdenum < 0.01 mg·L$^{-1}$; sodium < 0.02 mg·L$^{-1}$; nickel < 0.01 mg·L$^{-1}$) with an addition of 1M NaOH at a controlled temperature (40 ± 1 °C). The time duration of these tests was equal to 69 h. After 17, 24, 44, 52, and 69 h, the liquid in the coupon installation was checked. The parameter pH was measured using the multimeter CX-601 (Elmetron, Poland). In this study, two indices were used to identify the corrosiveness and scaling potentials of the tested water.

These include Langelier Saturation Index (LSI) and the Ryznar stability index (RSI) [12,13]. The LSI is an equilibrium model derived from the theoretical concept of saturation, indicating water saturation and its potential to precipitate calcium carbonate. This index considers the effects of calcium, total alkalinity, dissolved solids, and temperature. The RSI is the modification of LSI, and it offers better corrosion resistance (this parameter can withstand increased Ca hardness and pH values) [14]. The liquid from the coupon installation was also characterised using the inductively coupled plasma optical emission spectroscopy analysis (Agilent 5100 ICP-OES). This analysis was carried out following the PN-EN ISO 11885, PN-EN ISO 5667-3, PN-EN ISO 15587-1, and PN-EN ISO 15587-2 standards.

The corrosion rates (CR) were measured using the corrosion coupon weight loss measurements [15]. This parameter is defined as follows [12]:

$$CR = \frac{W_b - W_a}{A\,t} \tag{2}$$

where:

$W_b$—coupon weight measured before immersion in the water samples, g;
$W_a$—coupon weight measured after immersion in the water samples, g;
$A$—the exposed area of coupon, m$^2$;
$t$—exposure time, h.

The changes in the coupon structure were analysed using the SEM technique. In the test were used coupons produced by European Corrosion Supplies Limited, UK (see Table 3).

**Table 3.** Coupons used in the test of the coupon installation.

| Coupon Serial No | Material | Dimensions [in] | Weight |
|:---:|:---:|:---:|:---:|
| 919 | | | 10.6710 g |
| 920 | Mild steel | $3 \times 1/2 \times 1/16$ | 10.6579 g |
| 921 | | | 10.6779 g |
| 922 | | | 10.6979 g |

## 3. Results and Discussion

It should be noted that the Computational Fluid Dynamic (CFD) technique can provide access to qualitative and quantitative information concerning mixing performance in mixing devices [16]. The development of CFD models recently has contributed to the significant progress made in understanding fluid flow in various types of devices. The CFD technique is based on numerical methods that predict the governing transport mechanisms, and it can be used to describe many complex hydrodynamic phenomena [17]. It should be noted that the corrosion coupon orientation should be consistent to assess and compare different data sets.

Figure 5 shows the results of numerical simulations showing the velocity profiles in the tested coupon installation. The results showed that correct placement of the coupons in the zig-zag system is essential for corrosion testing. The coupons must not be placed in areas with strong swirls of fluid, e.g., at the flow through pipe elbows. According to the results from the numerical simulations, the coupons should be placed in such a way that the front of the coupon is directed in the opposite direction to the fluid flowing through the zig-zag system. Such an arrangement of the coupons allows the fluid to wash the coupon relatively homogeneously, which translates into a lack of areas where corrosion would have better conditions for its formation.

Numerical tests on determining the pressure decline in the tested installation were performed without and with coupons. Figure 6 shows the results of pressure drop changes versus the Reynolds number.

The applied coupons change the hydrodynamic conditions in the fluid flowing through the tested coupon installation. It should be noted that the hydrodynamic conditions in this setup are strongly dependent on the pressure drop. This parameter for both cases (installation with coupons and without the coupons) increases with the increment of the dimensionless Reynolds number, which ensures turbulences. At turbulent flow conditions (Re > 10,000), the pressure drop increases with the flow velocity (higher values of the dimensionless Reynolds number; this number varied from 14,910 to 30,660). As a result, in the pressure drop, that is, energy loss, the tested installation without the coupon has lower pressure drop values. As seen in Figure 6, placing the coupons in the system increases the pressure drop by an average of about 40%.

Figure 7 shows the changes in pH, LSI and RSI indicators. The pH fluctuated from 9.3 (for the initial time of the process) to 8.2 (after 69 h of time duration of the process). It should be noted that pH can hamper the process of scaling and corrosion [18,19]. The pH is also an essential factor influencing the formation of deposits such as $CaCO_3$ and $Mg(OH)_2$. This parameter is also the base for calculating LSI and RSI [4]. The LSI is allowed to assess the corrosivity of a water sample to dissolve or precipitate $CaCO_3$ (it is the main factor of water corrosivity). The positive values of LSI depict that water deposits $CaCO_3$ on the metal surface (e.g., coupons), indicating the state leading to the marginal level of corrosion. The negative value of LSI shows the corrosive nature of water. This parameter indicates the potential of scale formation in the cooling systems. In the case of these investigations, the LSI varied in the range between −0.61 (for the initial time of process) to −2.0 (after 69 h of time duration of the process). Generally, the LSI value above 0 is connected with the super-saturated water tendency (precipitates $CaCO_3$). The LSI value below 0 is characteristic of undersaturated conditions in water (dissolves $CaCO_3$). The LSI equal to 0 means we have saturated conditions ($CaCO_3$ in equilibrium). The RSI is the modification of LSI, and it was developed to assess water-scaling tendencies. A value of this index above 8.5 means that the water is highly corrosive. Moreover, the RSI value above 9.0 defines very intense dissolving of scale and corrosion [5]. According to the obtained results (see Figure 7), the coupon installation applied in this water test is not scale-causing and is highly aggressive.

Re = 14,910

Re = 27,090

Re = 30,660

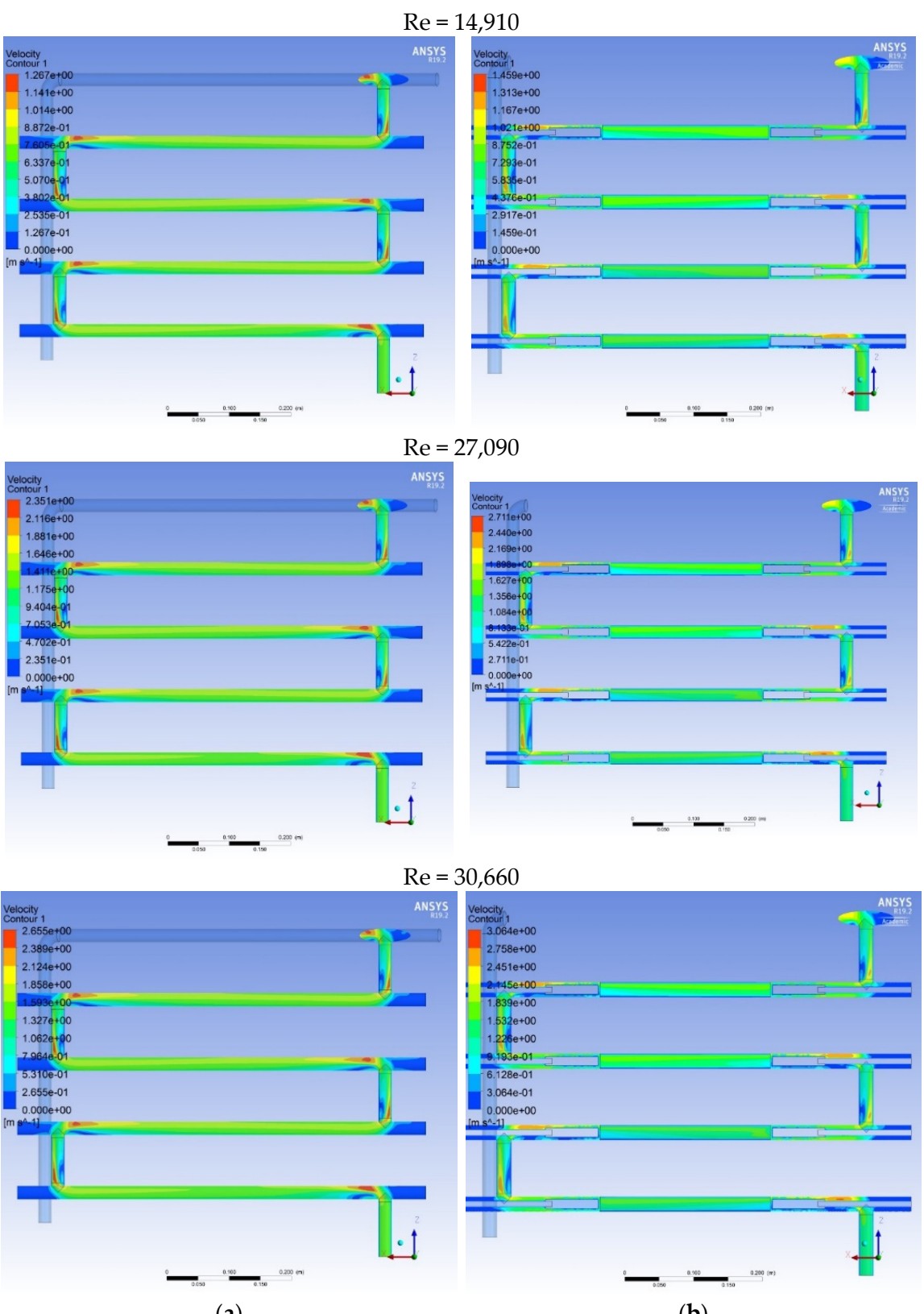

(**a**)　　　　　　　　　　　　　　　　　　　　　(**b**)

**Figure 5.** The results of numerical simulations (velocity profiles) for installation without (**a**) and with coupons (**b**).

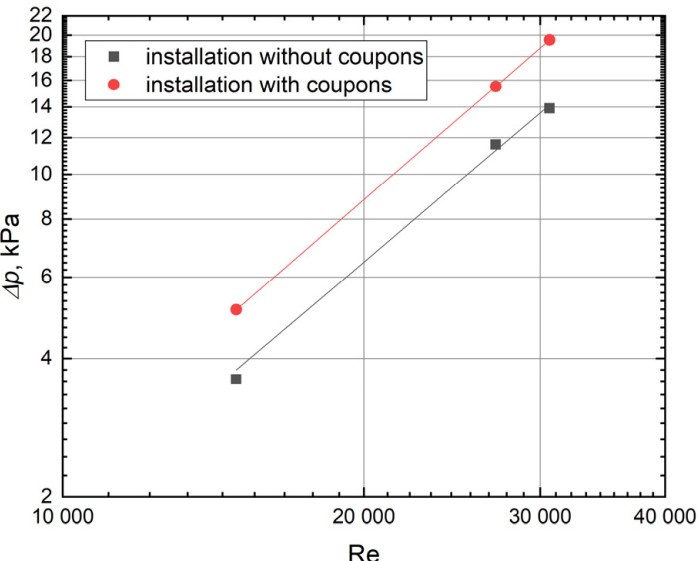

**Figure 6.** The relation between the pressure drop and the dimensionless Reynolds number for the tested installation.

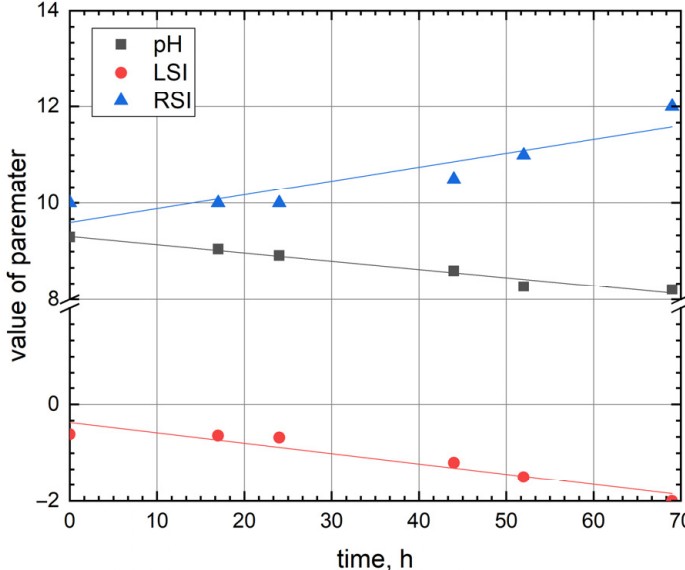

**Figure 7.** Time variation of pH, LSI, and RSI values for the tested coupon installation.

It should be noted that water's corrosivity and alkalinity have an inverse relationship. The rate of iron corrosion is low when alkalinity is high. This means that the increase in Ca concentration is connected with the decrease in water corrosivity [4]. The main reason for this is the formation of Ca scales of $CaCO_3$, which form a protective coat on the metal surface to regulate corrosion. The primary factors influencing corrosion in water systems are water temperature, pH (the impact of this parameter on corrosion indices is the most), Ca hardness, total dissolved solids (TDS), and alkalinity. The order of sensitivity in which water parameters affect corrosion indices is as follows: pH > alkalinity > Ca hardness > temperature > TDS [f].

Table 4 shows the elements content for the selected water samples from the coupon installation. The ICP-OES analysis allowed us to define the amount of iron leached into the medium. In the present study, the iron ion concentration increased by increasing the time duration of the process. When Ca concentration increases, water corrosivity decreases, since Ca forms scales of $CaCO_3$, forming a protective coat on the coupon surface to regulate

corrosion [4]. The obtained results give a clearer picture that the water corrosivity increases (Ca concentration decreases during the process by approx. 27%). Table 4 also shows changes in conetents over time for other elements (these results were obtained with the alpiaction of ICP-OES analysis).

**Table 4.** The element's contents (ppm) for the corrosion process are carried out using coupon installation.

| Metal | After 24 h | After 52 h | After 69 h |
|---|---|---|---|
| Ag | 0 | 0 | 0 |
| Al | 0.019 | 0.0059 | 0.0041 |
| Ba | 0.0003 | 0.0002 | 0.0002 |
| Ca | 0.8859 | 0.59 | 0.6467 |
| Cd | 0 | 0.0003 | 0.0001 |
| Cr | 0.0004 | 0.0004 | 0.0004 |
| Cu | 0.0249 | 0.0176 | 0.017 |
| Fe | 0.2875 | 0.3753 | 0.3967 |
| K | 0.0547 | 0.1114 | 0.1644 |
| Mg | 0.0435 | 0.0239 | 0.028 |
| Mn | 0.0029 | 0.0047 | 0.0048 |
| Na | 7.989 | 8.1361 | 8.1127 |
| Ni | 0.0017 | 0.0024 | 0.0027 |
| P | 0.0026 | 0.0072 | 0.0053 |
| Pb | 0.0027 | 0.0019 | 0.0016 |
| S | 0.407 | 0.2467 | 0.2623 |
| Si | 0.19 | 0.1433 | 0.1466 |
| Zn | 0.0189 | 0.0108 | 0.0106 |

Figure 8 illustrates the coupons exposed to the aggressive conditions obtained in the tested coupon installation. At the end of the test (after 69 h), observations of the coupons before cleaning were recorded (see Figure 8a). The samples were cleaned to remove all deposits and corrosion products (see Figure 8b). After cleaning, the coupons were weighted, and the CR (Equation (2)) was calculated from the weight loss.

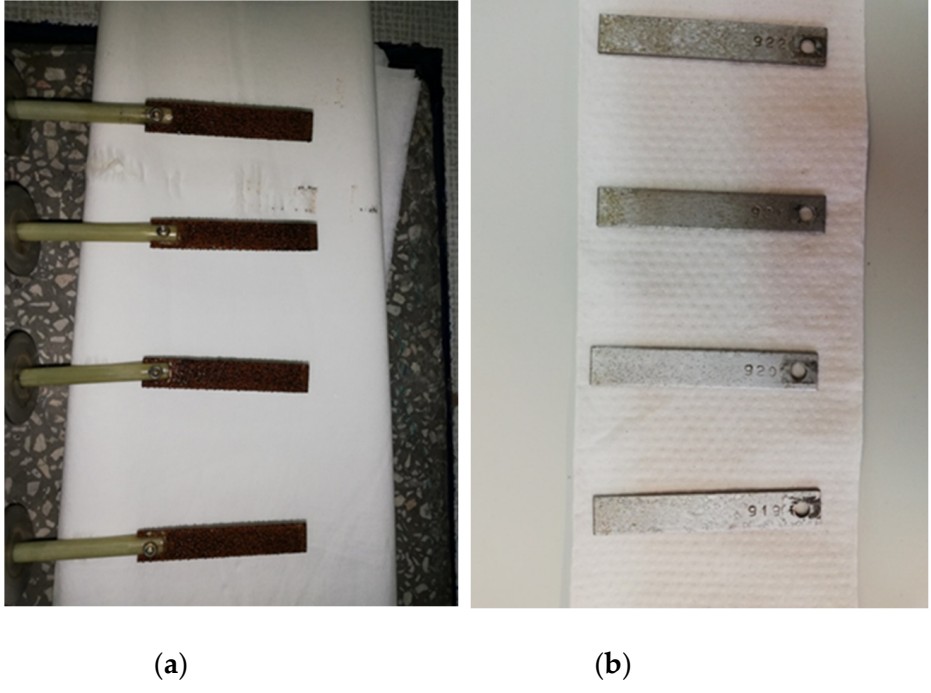

(**a**)        (**b**)

**Figure 8.** The view of coupons after the corrosion process in the tested installation: (**a**) coupons with all deposits and corrosion products; (**b**) coupons after cleaning.

The corrosion rate (CR) calculation for the tested coupons is collected in Table 5. The obtained CR values are changed from 1.8779 to 1.9706 $g \cdot m^{-2} \cdot h^{-1}$. It should be noted that the corrosion suffered by mild steel coupons was mainly due to a general type of attack.

**Table 5.** The calculated values of CR (Equation (2)) for the tested coupons ($W_b$ is taken from Table 3; exposed area of coupon, A, is equal to 0.002032 $m^2$; exposure time, t, is equal to 69 h).

| Coupon Serial No | Weight after Process $W_a$ [g] | $W_b - W_a$ [g] | $CR$ [$g \cdot m^{-2} \cdot h^{-1}$] |
|---|---|---|---|
| 919 | 10.4077 | 0.2633 | 1.8779 |
| 920 | 10.3914 | 0.2665 | 1.9007 |
| 921 | 10.4016 | 0.2763 | 1.9706 |
| 922 | 10.4250 | 0.2726 | 1.9443 |

The morphology of the coupons was investigated by scanning electron microscopy (TESCAN, VEGA SBU3) under 30 kV acceleration voltage. The typical SEM images of the used coupons are shown in Figure 9. The magnified images show that the surface of the coupons is violated by the presence of samples in an aggressive environment. These images show clear corrosion pitting caused by fluid flowing along the coupon in the tested installation.

Raman's spectra of the used coupons were recorded on a Raman InVia Microscope (Renishaw, Gloucestershire, UK) using a 785 nm laser. The analysed area was investigated in 100 points, with three repeats for each of them. Figure 10 presents three out of one hundren representative Raman spectra, performed during mapping, for each mild steel coupon of 919, 920, 921, and 922, correspondingly (A–D), its optical images (A′–D′) and respective Raman mappings of the intensity of the peak at 247 $cm^{-1}$ (A″–D″) performed with a laser length of 785 nm. The sample which was visually the most corroded according to the Raman spectrum (Figure 10E) was taken, and it shows peaks at 223, 247, 310, 377, 435, 513, and 674 $cm^{-1}$, which can be directly attributed to the $\gamma$-FeOOH (one of the main corrosion products) [20–22]. The most intensive one is at 247 $cm^{-1}$; therefore, the intensity variation of this peak has been observed during mapping studies. The red areas in the Raman mapping (Figure 10A″–D″) correspond to the dark areas presented in optical images (Figure 10A′–D′). The brighter red areas in the mappings are related to the most intensive $\gamma$-FeOOH content, indicating the areas where the corrosion was the most efficient. Analysing the data, it is clear that the corrosion process took place; however, it is inconsistent across the entire sample surface.

The current study found that the proposed coupon installation might be used to test the corrosion process. The apparatus used for these tests in the liquids or solutions was equipped with a temperature-stabilising system to maintain and control the assumed process temperature. The coupons were utterly submerged in corrosive water. In the case of the accelerated laboratory tests, one or more corrosive factors are strengthened (e.g., temperature, pH, relative humidity, moisture condensation, and concentration of corrosive components such as $SO_2$, $H_2S$, ammonia, acids, and chlorides). The corrosion process under these conditions is faster than the process carried out in operating conditions. In the case of this experimental work, the novel type of coupon installation was tested using the modified pH conditions (1M NaOH was applied to test the corrosion process with the application of the proposed setup). As mentioned above, the impact of pH on corrosion process indices is the most.

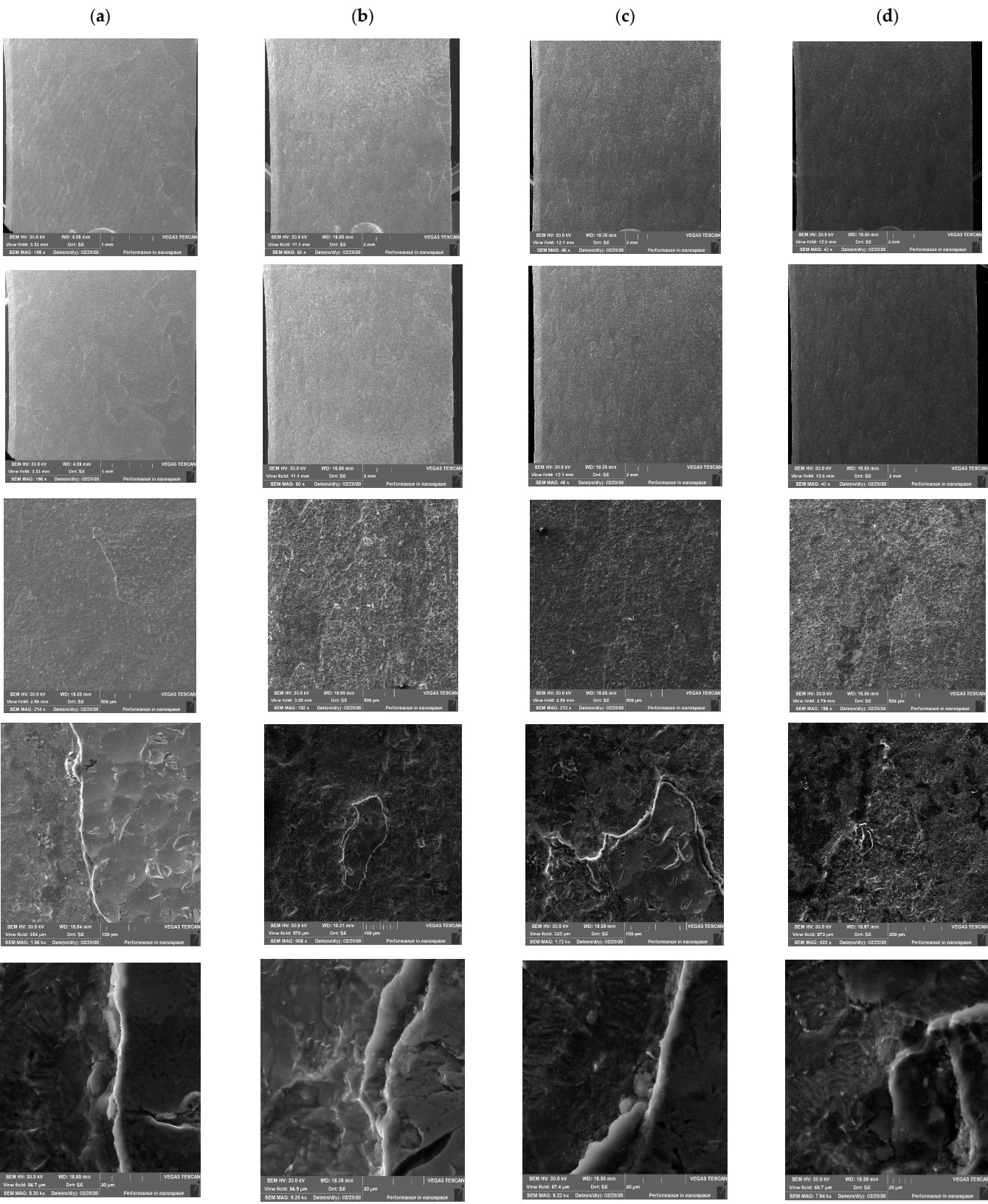

**Figure 9.** The SEM images of the tested coupons: coupon No 919 (**a**), coupon No 920 (**b**), coupon No 921 (**c**), and coupon No 922 (**d**).

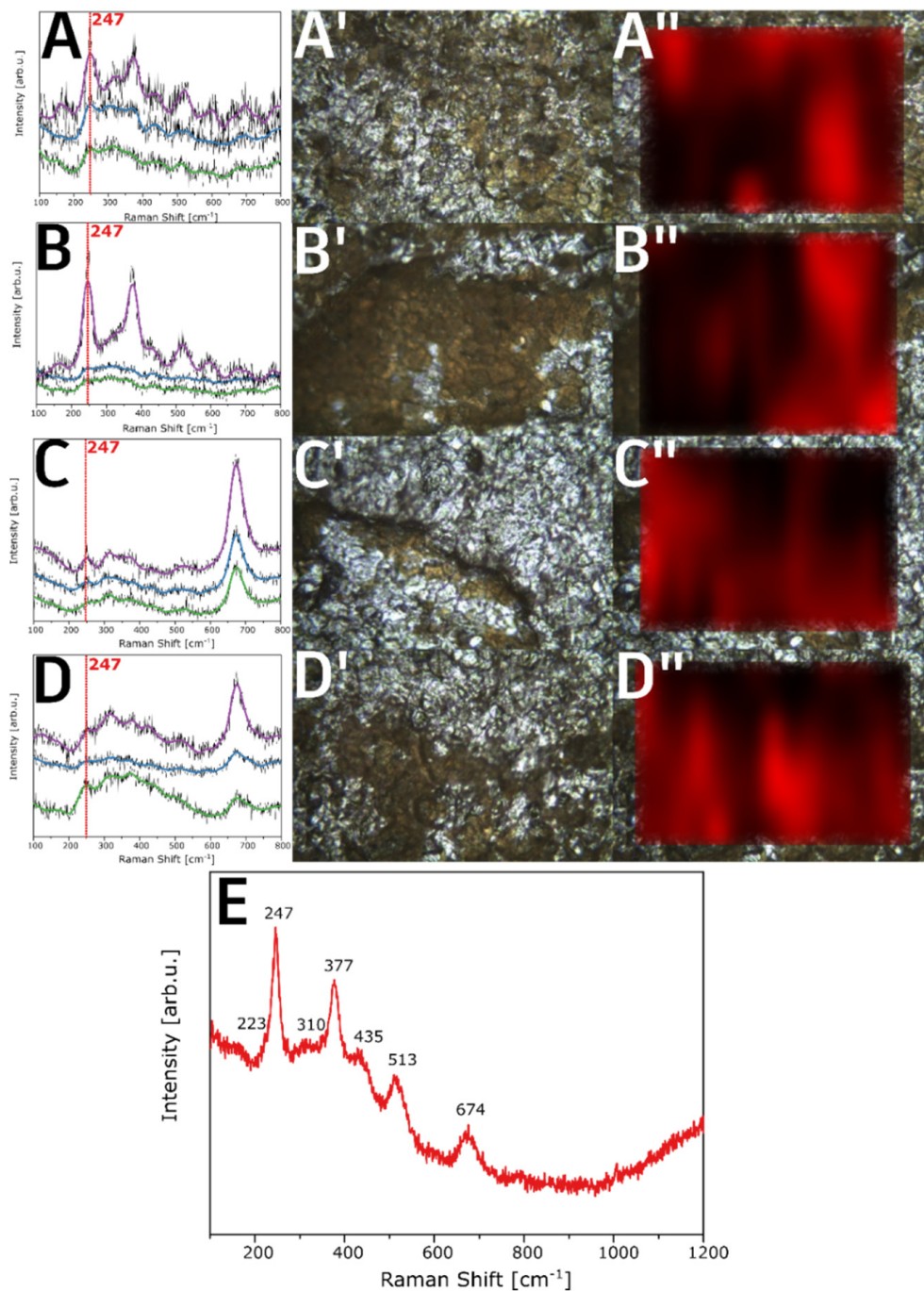

**Figure 10.** Raman mapping (785 nm) for 919 (**A**,**A′**,**A″**), 920 (**B**,**B′**,**B″**), 921 (**C**,**C′**,**C″**), 922 (**D**,**D′**,**D″**) and Raman spectra of corroded area (**E**).

It should be emphasised that the proposed coupon installation might be used in testing substances that can be treated as corrosion inhibitors (such tests are planned and will be performed for a new group of chemical substances). For example, the chemical treatment of boiler feed water might be controlled by the addition of ammonia to alkalising amines [8]. The alkaline environment is maintained to counter acidic conditions, where rapid corrosion is promoted by deteriorating the natural protective oxide layer of the metal. The reducing environment leads to generating a layer of porous magnetite ($Fe_3O_4$) that promotes flow-accelerated corrosion (FAC) [23]. To increase the FAC resistance, oxygen is added to the water [24,25]. This allows the generation of the protective and less soluble layer of FeOOH or $Fe_2O_3$ (the application of these layers is connected with the deterioration

of the entire steam-water cycle persformance). From a practical perspective, the corrosion process might be limited by the application of phosphates [26] or film-forming amines (FFA) [27–29]. It should also be noted that the new types of corrosion inhibitors were tested based on organic and organometallic compounds [30]. Moreover, polymeric corrosion inhibitors based on ionic coumarate groups were applied [31].

## 4. Conclusions

From the results of the present study, the developed and constructed coupon installation might be successfully used in testing corrosion phenomena. The tests of this installation were carried out with the application of the controlled conditions (demi water with the addition of 1M NaOH). The process conditions were based on the computation of the different corrosivity indices (LSI and RSI). The liquid samples from the tested coupon installation were also characterised using ICP-OES analysis. The obtained results were allowed to define the amount of iron and calcium leached into the medium. It should be emphasised that the numerical analysis of the tested installation showed that the placement of the coupons in the installation could be critical for corrosion testing. In the case of these investigations, the corrosion process was evaluated using the corrosion rate parameters and SEM images. The obtained Raman spectra for the used coupons showed that the corrosion process took place; however, it is inconsistent across the entire sample surface. Probable differences in the corrosion process might be due to possible disturbances in the fluid flow near the coupon surface. It should be noted that the tested installation is allowed to monitor the corrosion process and observe the progress of corrosion on the surface of the coupons. The consequences of the corrosion process are many and varied, and can lead to infrastructural and economic damage. Determinants, such as corrosion and scaling, should be considered when planning various types of apparatuses or conditions applied in many branches of industry. It was deduced from the present study that the tested conditions can cause damage to the steel elements of systems and can render financial losses to the industry. Hence, it is necessary to consider various corrosion indices and plan the maintenance procedures accordingly. The corrosion process might be curbed by adjusting pH and employing inhibitors to inhibit chemical reactions (e.g., silicates or polyphosphates). A vital strength of the present study was the development and testing of a novel coupon installation. The proposed apparatus can be successfully used in experimental works using aggressive environments and for testing new compounds with the potential of inhibitors.

## 5. Patents

The coupon installation presented in this paper is the subject of patent application N P.440109; WIPO ST 10/C PL440109.

**Author Contributions:** Conceptualisation, R.R. and D.M.; methodology, R.R. and D.M.; software, R.R. and M.K; validation, R.R., M.K. and D.M.; formal analysis, R.R.; investigation, R.R., D.M. and K.W.; resources, R.R. and M.S.-W.; data curation, R.R. and M.S.-W.; writing—original draft preparation, R.R.; writing—review and editing, R.R.; visualisation, R.R. and M.K.; supervision, R.R.; project administration, R.R., M.S.-W. and D.M.; funding acquisition, R.R., M.S.-W. and D.M. All authors have read and agreed to the published version of the manuscript.

**Funding:** This research was funded by European Union from the European Regional Development Fund under the Regional Operational Programme of the West Pomeranian Voivodeship 2014–2020; grant number RPZP.01.01.00-32-0013/18 (Operation 1.1: Research and development projects of enterprises; Project type 2: Research and development projects of enterprises with preparation for implementation in business activities; Project title: Industrial research and development work on an innovative application of amine derivatives and polymer compounds for the production of innovative preparations for water conditioning in industrial boiler and cooling systems).

**Data Availability Statement:** Data sharing not applicable.

**Acknowledgments:** Authors want to acknowledge the Department of Nanomaterials Physicochemistry, the West Pomeranian University of Technology in Szczecin (SEM images and Raman analysis).

**Conflicts of Interest:** The authors declare no conflict of interest.

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
