# Peer review of "Analysis of the Corrosion Process with the Application of the Novel Type of Coupon Installation"

_processes, doi:10.3390/pr10122468_

Round 1
Reviewer 1 Report
Dear authors
This is an interesting research related to analysis of the corrosion process. However, it should be revised according to several major problems before publishing
1. Figure 1 and 2 should be integrated and some parts or components Figure 2 should be marked and named
2. In Section 3, the authors simply described results. It lacks dept discussions. Please added some discussions as well as change the section title to " Results and Discussions"
3. Table 4 and 7 should be changed to Figures.
Author Response
We would like to thank you very much for the valuable comments concerning our manuscript. We would like also to thank all reviewers very much for their effort in the revision of the manuscript. All suggestions were strong taken into consideration what surely has improved the value of the paper which reports, we believe, a very important message. All changes are marked by the red color font.
- Figure 1 and 2 should be integrated and some parts or components Figure 2 should be marked and named
I appreciate for all the comments which are positively formulated for improving the quality of the manuscript. Thank you for the valuable efforts and time from all the Reviewers. Your careful attention to the manuscript was a great aid to the improvement of this paper and my future studies. Thank you for pointing out all the mistakes in my work. The manuscript is revised and the corrected version of the paper includes information in order to clarify the errors.
We agree with the reviewer, Figure 1 and 2 have been integrated.
- In Section 3, the authors simply described results. It lacks dept discussions. Please added some discussions as well as change the section title to " Results and Discussions"
Thank you for comment. We added some discussion connected with the obtained results (see red marked text in this chapter).
- Table 4 and 7 should be changed to Figures.
We agree with the reviewer, Table 4 and 7 have been changed to figures.
Reviewer 2 Report
The authors propose a novel type of coupon installation which might be used to measure and forecast the corrosion process. The topic is interesting and the manuscript can be published after a minor revision. There are a number of points that authors should take into account.
i. Lines 253-254. Authors write: "It should be noticed that the corrosion rate can be estimated by employing this method and this analysis might be treated as a kind of in situ method for monitoring corrosion [23]" Why didn't the authors present this calculation and compare the obtained results with the results of the weight loss method?
ii. Lines 115-117: Authors write: "It should be noticed that these coupons might be placed into the working liquid parallel to the direction of fluid flow. In addition, it is possible to monitor the corrosion process taking place on the coupons directed in and against the direction of the fluid flow." However, the data corresponding to the different arrangement of coupons are missing in the manuscript.
iii. Lines 274-275: Authors write: "The corrosion rate of coupons varied and changed with the placement of the coupons in the tested installation, but the obtained values of the CR are similar." Do the authors mean to say that each of the 4 coupons was located differently in the installation? And do the authors draw conclusions from data for only one sample in each particular location? It is not correct. Multiple duplicates are required in each case. Must have stats.
iv. Lines 256-257: Authors write: "It should be noticed that the obtained analysis is allowed to assess the release of heavy metals connected with the corrosion process (eg Fe, Cu, Cd, Pb, Mn, and Cr)" Where did all these metals come from in the solution and why did the concentration of some of them decrease over time (Al, Ba, Cu, Mg, Pb, S, Si, Zn), and some increase (Cd, K, Mn, Na, Ni, P) .(Table 5).
v. Line 151. CFD needs to be decoded.
vi. Line 182. The authors should give the composition of demi water.
vii. Table 3. Specify the units in which the dimensions are given.
viii. Table 7. The caption under the picture does not match it

Author Response
We would like to thank you very much for the valuable comments concerning our manuscript. We would like also to thank all reviewers very much for their effort in the revision of the manuscript. All suggestions were strong taken into consideration what surely has improved the value of the paper which reports, we believe, a very important message. Please find the attached file.

Reviewer 3 Report
The author can refer to the revision comments to further revise the article to improve the quality of the article:
1.The abstract is incomplete and lacks generality, and the introduction to the main content of this article is insufficient
2.Introduction is too long. It is recommended to focus on the purpose and significance of this article
3.The conclusion is highly eventful, but lacks the exploration of multiple influencing factors of the result/experiment, and the experimental logic and process description are insufficient
4.The format of the chart title is inconsistent with the template requirements, for example: Table 7, etc
5.It is noted that your manuscript needs careful editing by someone with expertise in technical English editing paying particular attention to English grammar , spelling , and sentence structure so that the goals and results of the study are clear to the reader .
6. Water quality index are mentioned in the key words, but they are not specifically described in the text, especially in the introduction。
My opinion is minor revision.
Author Response
We would like to thank you very much for the valuable comments concerning our manuscript. We would like also to thank all reviewers very much for their effort in the revision of the manuscript. All suggestions were strong taken into consideration what surely has improved the value of the paper which reports, we believe, a very important message. All changes are marked by the red color font.

Round 2
Reviewer 1 Report
Yes, the manuscript was improved a lot. I think it can be accepted for publication